# On the development of a hardware-in-the-loop wind tunnel setup to study the aerodynamic response of floating offshore wind turbines

Federico Taruffi<sup>1</sup>, Shakthi Thinakaran<sup>1</sup>, and Axelle Viré<sup>1</sup>

<sup>1</sup>Faculty of Aerospace Engineering, Delft University of Technology, Kluyverweg 1, 2629 HS Delft, the Netherlands **Correspondence:** Federico Taruffi (F.Taruffi@tudelft.nl)

# Abstract.

In floating wind turbines, the met-ocean conditions lead to motions of the floater affecting the rotor aerodynamic loads, which in return influence the motion of the floater, in a highly coupled way. Numerical design tools have proven to fail to predict some aerodynamic phenomena, such as the increase in thrust variation caused by unsteady effects. Thus, experimental

- testing is essential for tuning and validating these codes. Hybrid testing in wind tunnels, by reproducing numerically and actuating the floater motions while measuring aerodynamic loads on a physical scale turbine model, overcomes the scaling issues of traditional wave basin tests allowing a higher fidelity in the reproduction of the aerodynamics. This work presents the development of a hybrid hardware-in-the-loop setup designed to study the aerodynamic response of floating wind turbines in wind tunnels. A scale model of a multi-megawatt floating wind turbine is mounted on top of a six degrees-of-freedom
- hexapod robot. The full coupling of aerodynamic and floater dynamics is obtained with a hardware-in-the-loop approach with force-feedback-motion-actuation architecture. The rotor loads measured on the physical rotor are fed into a floater dynamic numerical simulator which calculates the motion in real-time and actuates it through a moving platform called hexapod. Key outcomes include the development of a hardware-in-the-loop numerical model with a force correction method to cope with scaling effects and an assessment procedure to verify the simulator, correction model, and measurement-actuation chain. The
- aerodynamic effects on the motion response are preliminarily investigated on a 10MW floating concept, with direct estimation of the rotor aerodynamic damping showing a 210% increase of damping in pitch with the turbine in operation. The capability of testing combined wind and wave cases is also demonstrated, setting the framework for future studies.

# 1 Introduction

Offshore wind turbines are mostly installed as bottom-fixed structures rigidly mounted on the seabed or are moored to the seabed using floating support structures. In deeper waters exceeding 60 meters in depth, floating turbines anchored on the seabed are more economical (van Kuik et al., 2016). As suitable shallow-water sites become increasingly scarce, floating offshore wind turbines (FOWTs) offer a promising alternative, unlocking the potential to harness vast wind resources in deeper seas and opening new opportunities in countries without shallow seas.

FOWTs experience motions in six degrees-of-freedom (DOFs). The floating nature of the wind turbine makes it susceptible to various excitations, including changes in the met-ocean conditions, interactions between the wakes and the turbine, and the

operation of the turbine itself. The aerodynamic performance of FOWTs is affected by these motions, with the aerodynamic loads acting on the rotor being highly coupled with the floater dynamics. In some cases, unsteady aerodynamic effects are observed due to the interaction of the wind turbine blades with their own wake. These complex interactions are often not yet fully understood.

It is currently challenging to numerically predict the unsteady aerodynamic loads under certain conditions. In previous studies (Taruffi et al., 2024b) the results from an experimental test were compared with the quasi-static theory which failed to predict the unsteady aerodynamic loading at high frequency. Discrepancies in predicting these unsteady loads were also found with high-fidelity numerical models, such as large-eddy simulations with an actuator line model (Taruffi et al., 2024a). In the field, some recently-deployed FOWTs also needed premature maintenance on their rotors. This shows the need to better understand the unsteady phenomena and derive more accurate numerical tools for the design of FOWTs. 35

Experimental testing provides an effective way to investigate these complex phenomena and tune the numerical models. Full-scale testing or large-scale model testing can be done at sea. This was done by Viselli et al. (2015) with a 1:8 FOWT in the Gulf of Maine and by Ruzzo et al. (2021) with a 1:15 floating multi-purpose-platform including a wind turbine. However, this is expensive and the real offshore conditions are difficult to reproduce numerically. Hence laboratory tests with scaled models

are often preferred.

> Testing FOWTs is traditionally done in a wave basin with Froude scaling. This allows for accurate hydrodynamic load reproduction, enabling the understanding of the floater hydrodynamics. For example, Cermelli and Aubault (2010) studied the hydrodynamics of a 1:67 scaled FOWT in a wave basin with Froude scaling to validate a numerical model. In order to include aerodynamic loading, wind generators with thrust disks or geometrically scaled blades were used to emulate the aerodynamic

- thrust force on the rotor. These methods successfully generated an aerodynamic thrust force but did not match the values of 45 the full-scale model. The study by Goupee et al. (2014b) that involved testing a 5 MW wind turbine with different floaters in a wave basin with geometrically scaled blades revealed their drawbacks. This is due to the use of the Froude scaling law that does not allow for reproducing the aerodynamic forces correctly (Robertson et al., 2013). A Reynolds' similarity law does allow for accurate representation of the aerodynamics but it compromises the hydrodynamics and gravity related loads and
- also leads to impossibly high wind speeds. This scaling law conflict is unavoidable in fully-physical wave basin scale tests. To alleviate it, later studies followed a performance-based scaling methodology for the rotor by choosing an arbitrary velocity scaling factor for reproducing aerodynamic thrust more accurately while keeping the Froude scaling for the structure (Goupee et al., 2014a; Zhao et al., 2018; Doisenbant et al., 2018). However, this approach still has limitations. The wind generators only produce low-quality wind flows, and the correct mass scaling is not achievable in the turbine leading to an inaccurate centre of

gravity.

> Hybrid testing overcomes these challenges. Instead of physically applying both the aerodynamic and hydrodynamic loading, one part is physically tested while the other is numerically simulated, effectively eliminating the scaling conflict. This can be done in wave basins where aerodynamic thrust is simulated with the help of winches or thrusters, or in a wind tunnel where an actuated platform is used to behave as the floater. The choice depends on the scope of the study, with the wave basin and wind tunnel setups meant to perform studies on hydrodynamics and aerodynamics respectively. Due to the high level of

coupling between the aerodynamic loads acting on the rotor, and the motion response of the floating system, hybrid tests where the numerical part contribution is pre-calculated may be suitable for specific studies but are not suitable to truly reproduce realistic conditions that are necessary to investigate the aerodynamic response and the stability of a design solution. To allow this, hardware-in-the-loop (HIL) technology is used, where the numerical part is simulated in real-time and accounts for the real-time measurements from the physical model to emulate the missing physics.

The first approach for real-time hybrid experimental testing in wave basins was developed with a ducted fan actuator by Azcona et al. (2014). The speed of the fan was adjusted to emulate different aerodynamic thrust forces. This methodology was later used to study the non-linear hydrodynamic effects of a FOWT as described in Azcona et al. (2019) and it overtook the traditional non-hybrid testing methods as it was confirmed to have better accuracy. Other researchers such as Oguz et al. (2018), Armesto et al. (2018) and Thys et al. (2018); Bachynski et al. (2016) developed it further to include multiple fans or winch cables in order to better emulate aerodynamic forces. These testing methods were used to verify and/or compare FOWT

designs, observe complex interactions, or tune numerical codes.

The HIL setup in wave basins allowed for validating floater designs and studying the complex dynamics. However, to analyse aerodynamic effects such as unsteady aerodynamics and wake interactions, testing in a wind tunnel is required. Hybrid

- experimental testing in wind tunnels was pioneered by Belloli et al. (2020) who developed a HIL setup with a 10 MW thrust-scaled wind turbine model mounted on a two DOFs slide first (Bayati et al., 2017) and a six DOFs hexapod later (Bayati et al., 2018b). A numerical model was developed to solve the floater dynamics, but still with limited force feedback (Bayati et al., 2018a). Recently, a 15 MW FOWT was tested with a 1:100 length scale with the same setup (Fontanella et al., 2023b). Hybrid wind tunnel testing was also conducted by researchers such as Rockel et al. (2014) and Schliffke et al. (2020) to study the wake
- characteristics of FOWTs, however without hardware-in-loop coupling. They experimented with highly scaled models under prescribed motions in a boundary layer wind tunnel. Overall, research that used hybrid testing in wind tunnels to study FOWT dynamics is scarce.

In a previous work by Taruffi et al. (2024b), a setup was developed at Delft University of Technology for hybrid testing of FOWT in wind tunnel using a 6 DOFs hexapod robot. While that study used imposed motions, in this work the setup is further developed to investigate the aerodynamic response of FOWTs using HIL technology. The setup comprises physical aerodynamic loads that are fully coupled with numerically simulated floater dynamics. A numerical model is developed that solves the dynamics of the floater in 6 DOFs which includes correcting the measured force due to non-Froude scaling and simulating the waves. Due to the scaling mismatch, the reliability of the force correction plays an important role in the accuracy

of the setup for simulating real-world dynamics. This work primarily focuses on the verification and validation of the setup,

especially the force correction methodology, to ensure its accuracy to be utilised in future studies. An objective is to make the setup versatile enough to accommodate various FOWT designs, enabling the testing of larger turbines and different floater configurations with ease. Preliminary free decay and combined wind and wave tests are performed on a 10MW FOWT concept, investigating the aerodynamic damping effect on the motion response and assessing the capability of the setup to reproduce realistic wind and wave conditions.

# 95 2 The hybrid setup

The hybrid setup comprises a wind turbine scale model - the physical subsystem - and a six degrees-of-freedom hexapod robot - the numerical subsystem. The skeleton of the setup is the same as the previous work (Taruffi et al., 2024b) which focused on the study of rotor unsteady aerodynamics. In addition to the previous setup, a real-time controller bridges the two subsystems to implement the hardware-in-the-loop functionality. This setup is for use in wind tunnels. For this work, it was tested in the Open Jet Facility (OJF) at the Delft University of Technology, a closed-loop open jet test section facility with a 2.85m × 2.85m nozzle opening into a 13m long, 8m high open test section. At 1m downstream the nozzle, where the rotor is placed, the flow is uniform with a turbulence intensity of 0.5%. A view of the hybrid setup in OJF, with its components highlighted, is shown in Fig. 1.