# Peer review of "On the development of a hardware-in-the-loop wind tunnel setup to study the aerodynamic response of floating offshore wind turbines"

_Wind Energy Science, 2025_

## Author Comment (AC1)

**Authors' comment**

Dear Referees,

Thank you for your insightful feedback. In this document, you will find the answers to your questions and comments, and a description of what will be modified in the revised manuscript.

To improve the soundness of the assessment of the setup, which is the core of the work, the revised manuscript will include additional analyses, with a more thorough verification of the aerodynamic force estimation and additional comparisons between the floater numerical model and the benchmark.

With kind regards,

The authors

**Reviewer 1**

Dear Authors,

I was invited by the Associate Editor to review your article and was pleased to accept, as the topic is of significant interest to me. I dedicated a considerable amount of time to its review. As you state, there are currently very few setups capable of conducting hybrid wind tunnel experiments for floating wind turbines and the development of such systems could be important for advancing floating wind technology. Therefore, I believe the topic can be of interest to the research community and the readers of this journal.

However, the manuscript requires substantial revisions to be technically sound, reproducible, and impactful.

As it stands, the article primarily describes the development of a HIL setup as a preparatory step for your future experiments, with limited contribution to the understanding of coupled dynamics in floating wind turbines. While I fully understand the need to establish a foundation for future work, a publication must provide clear value and insights for the reader.

**General comments**

I recommend that you clearly state, in the introduction, the relevance of your work, its main objectives, and its novelty compared to previous studies. In other words, the readers should understand what they can gain from the article. To support this, I suggest expanding the literature review so that gaps in the current state of the art become more evident, and your contribution can be better contextualized.

The relevance of this work lies in the importance of having experimental data about floating wind turbines while relying on a few setups capable of doing so. Its main objective is to show the development and assessment of an HIL experimental framework for future wind tunnel studies targeted at the aerodynamic response. In this, assessing the aerodynamic force estimation method is a crucial point for such HIL setups. Here lies also the key novelty of this setup, by using direct acceleration measurements to estimate the aerodynamic contribution from the rotor loads measured by the loadcell. This will be highlighted in the introduction, and the literature review will be expanded, also following the comments of Reviewer 2.

A second general point concerns the technical accuracy in some parts of the manuscript. I believe responding to the Specific Comments can help improve this aspect, but I also encourage greater care in ensuring accuracy and clarity in the article.

**Specific comments**

- Abstract: "feedback control". Which kind of control? Generator speed/power?
  We couldn't find this part in the abstract.
- Abstract: "Numerical design tools have proven to fail... by unsteady effects". Relatively recent research and community efforts have shown that numerical models are quite effective in capturing the thrust response of floating wind turbines especially without active rotor speed and blade pitch control as in the present work. I ask the authors to be

clearer on what the numerical codes fail to predict or remove this kind of statements if they are not true or not meaningful for the present work.

This statement is elaborated in the introduction. It refers to a couple of studies from the authors (https://doi.org/10.5194/wes-9-343-2024, https://doi.org/10.1088/1742-6596/2767/6/062010) where experimental results of thrust variation have been compared with the Quasi-Steady Theory and LES simulation with Actuator Line Method. The sentence in the abstract will be modified to be less affirmative.

- Abstract: rephrase the sentence "by reproducing … scale turbine model" because it's not clear.

It will be changed in the manuscript.

- Abstract: "higher fidelity" with respect to what?

It refers to wave basin tests with physical scale rotors

- Abstract: "with direct estimation … with the turbine in operation". Is this damping increment found also in simulations? Was it expected?

No numerical simulations of decays in wind conditions were performed in this study, so no direct comparison is possible. Numerical tools (FAST) were only used to assess the floater dynamics part, while the aerodynamic part is physical. A comparison of the aerodynamic response with mid- and high-fidelity numerical models could be a follow-up study. An increase in damping of pitch, surge and yaw DOFs is expected, and this setup allows for quantifying it with measured aerodynamic loads.

- L28: after "due to the interaction of the wind turbine blades with their own wake" you should cite work on this topic. Examples are [1, 2] but feel free to cite others.

Citation of [1,2] will be included in the manuscript.

- L31-34: "the results … numerical models". How likely it is that the unsteadiness is due to the wind turbine aerodynamics and not something related to the experimental setup (like bending of blades, tower, load cell flexibility and related resonances)?

These options were carefully considered by the authors in that work [cite]. To sum those considerations: flexibility of tower and load cell are taken into account by using the tower-top acceleration signal to normalise the loads and by comparing the same motion case in no-wind and wind conditions. The first blade frequency is around 16Hz, far from the higher motion frequency tested of 5Hz.

- L37-38: "This was done … a wind turbine". Did they use experimental results to investigate aerodynamics modeling? If not, maybe the citations are not relevant here.

The aerodynamic modelling of the large-scale wind turbine model mounted on the multi-purpose platform in https://doi.org/10.1016/j.apor.2020.102487 was investigated in https://doi.org/10.5194/wes-8-71-2023. There, the performance scaling was assessed with field measurements. Overall, we believe that citing, briefly, other types of experimental testing for FOWTs is relevant for the storyline.

- L48: "A Reynolds' similarity law…" Is Reynolds scaling (i.e., keeping the Reynolds constant) even possible?

No, it's not possible at lab scales, as stated with "and also leads to impossibly high wind speeds". To improve clarity, this will be changed in the manuscript.

- L62: "for specific studies". Can you be more specific about the kind of studies where prescribed motions are useful?

Prescribed sinusoidal motions are useful for studies about e.g. unsteady loads and wake measurements, where a high number of cycles is preferred (e.g. using PIV to measure the wake, only few seconds can be recorded so a more complex and longer

time history of realistic motion couldn't be recorded entirely; sinusoidal motion with different dofs, frequencies and amplitudes allow to understand better the effect of the motion characteristic on the wake). Prescribed realistic motions, like motion output from simulations (OpenFAST) are useful if the two-way interaction is not the object of study or regarded as not important and the focus is on the motion-induced rotor loads or power.

- L72: "complex interactions" interactions of what?
  It refers to the highly coupled dynamics of FOWTs, e.g. the interaction between the aerodynamic and hydrodynamic or mooring loads, or the interaction of the turbine control system with the stability of the floater.

- L79-81: "Hybrid wind tunnel testing … wind tunnel". I'd say that these references are not very meaningful here. There many studies of wind turbine aerodynamics / wake using prescribed motions, and the two cited by the authors are some of them. However, the rest of the discussion is on HIL experiments, and I think it would be better to focus on those.
  The citations will be removed from the manuscript to focus on HIL.

- L87: "correcting the measured force". At this point of the article, it is not clear what force correction is. I would say something like "estimating rotor loads from the available measurements".
  This will be changed accordingly in the manuscript.

- L109: Move "The relevant properties are reported in Table 1" before the sentence "A complete …".
  This will be changed accordingly in the manuscript.

- Table 1: Make length scale and velocity scale consistent with what is reported in the text body.
  This will be changed accordingly in the manuscript.

- Table 1: replace "Tilt" with "Shaft tilt".
  This will be changed accordingly in the manuscript.

- Figure 2: in the caption it is said "The experimental points are averaged over two sets." Two sets of what? Please explain it in the text. You should briefly explain how the characterization was performed. I get that thrust is the main driver of the floating wind turbine response and it was the main objective of the scaling but also torque is important because the turbine is in the end a power generating machine. Can you also show the torque corresponding to the measured thrust reported in Fig. 2?
  The characterisation of the rotor performance was performed in https://doi.org/10.5194/wes-9-343-2024 where Figure 4 shows the thrust and power curve. Figure 2 of the current manuscript averages the values of the two sets of tests that are reported in Figure 4. Each set consisted of static tests (no motion) for 5 below-rated wind speeds and corresponding rotational speeds (TSR of 7.5). About 60 seconds were recorded for each test, and the forces in x-direction (thrust), moment in x-direction (torque) and rotor speed from encoder were averaged and power computed. In the current manuscript, we decided to show only the thrust as an indication of the rotor performance most relevant to the study. However, the paper showing both thrust and torque characterisation is cited in the caption of Figure 2 of the current manuscript. The torque curve is attached here. The torque values don't match the full-scale concept, resulting in sensibly lower values. This was expected and also shown in the paper about

the aerodynamic design of the turbine (by Fontanella et al.).

[Figure]

- Figure 2: the figure displays only thrust. Can you show the corresponding values of torque?
  See point above.

- L115: change "in the previous study (Taruffi et al., 2024b)." with "in the previous study of Taruffi et al., 2024b."
  This will be changed accordingly in the manuscript.

- L117: "without tracking errors" How this relates to amplitude of motion? At least you have a limit on the amplitude imposed by the maximum acceleration of 1g.
  The limitation is given by maximum velocity and maximum acceleration. In https://doi.org/10.5194/wes-9-343-2024, the dotted line in Figure 3 shows the limit in terms of amplitude and frequency for surge motion.

- L118: "most load conditions" Is it true? A wave period of 8s, which is not so uncommon corresponds to a mode-scale frequency of 6.3Hz which is above the range where the hexapod works ok. The maximum model scale acceleration of 9 m/s2 corresponds to a full-scale acceleration of 0.54 m/s2 (0.06g) which seems quite less than what a floating wind turbine can experience. Can you be more accurate on what you mean with "most load conditions"? Maybe it is more accurate to say something like "load conditions with mild waves" etc.
  The hexapod can reproduce the motion induced by most "operational" sea states. The operational wave cases that were considered are in the table below, and the hexapod was capable to track the motion of cases 1 to 5, while cases 6 and 7 where above its limits. FAST simulations have been performed for each wave case (no-wind conditions) to assess that the motions are within the actuation limits. In the text, normal,

operational or mild will be stated.

| Operational 1 | 5400 | Irregular H$_s$=1.38 m, T$_p$=7.0 s |
|---|---|---|
| Operational 2 | 5400 | Irregular H$_s$=1.67 m, T$_p$=8.0 s |
| Operational 3 | 5400 | Irregular H$_s$=2.20 m, T$_p$=8.0 s |
| Operational 4 | 5400 | Irregular H$_s$=3.04 m, T$_p$=9.5 s |
| Operational 5 | 5400 | Irregular H$_s$=4.29 m, T$_p$=10.0 s |
| Operational 6 | 5400 | Irregular H$_s$=6.20 m, T$_p$=12.5 s |
| Operational 7 | 5400 | Irregular H$_s$=8.31 m, T$_p$=12.0 s |

- L124: "rotor force correction process". Again, I think it's not clear here what you mean with correction.

  This will be changed in the manuscript, consistently with the modification requested above.

- L141: "NREL 5MW ... to the scale". Add reference documents for the two wind turbine designs and be more accurate on what you mean with adjusting the operating parameters (rated wind speed, working TSR) and why this is acceptable (e.g. the optimal TSR of the IEA 15MW is different from the one of the DTU 10MW, thus if you run your rotor at the TSR of the IEA 15MW the local angle of attack is suboptimal for the rotor design).

  We agree with the observation. We preliminary did a OpenFAST analysis of our scale model, and we reproduced the 15MW thrust curve with about 5% error at rated with respect to the OpenFAST of the 15MW, which could be consider acceptable. This has not yet been tested experimentally. However, the text will be modified, stating that it is possible to use the turbine to reproduce different rotor ratings, without specifying the IEA 15MW or NREL 5MW.

- Table 3: Again, be consistent with the text and other tables when reporting the scale factors. Moreover, the first two lines of Table 3 are repeated from Table 1. I suggest removing them from Table 1.

  This will be changed accordingly in the manuscript.

- Section 3: Which is the delay in the measurement-actuation chain? Where it mostly comes from? (e.g. the hexapod motion controller). Which is the transfer function between motion setpoint and actual motion of the hexapod?

  An estimation of the delay is given in Section 3.3.3. The Bode diagram between the position setpoint and actual position (of surge DOF) evaluated for the wind and wave case in Figure 9 of the preprint is reported here (the transfer function is estimated with 2 poles to represent the mechanical system, and the fit is reported after). The time history (cut) and PSD of the signals are also reported below. A time history showing setpoint and actual position and the Bode diagram will be included in the manuscript in the section about the hexapod.

[Figure]

- L174-178: "Differently from Belloli ... are not measured". The same is done in [3] which is the development of Belloli 2020, for the same reason mentioned here. It is also said "We preferred to have the load cell at tower-top rather than at tower base (like in Belloli et al. 2020) because in this way the aerodynamic components are larger fractions of the signal, and to avoid introducing deformability at tower base that would reduce the frequency of the turbine first fore-aft mode."
This statement will be changed in the manuscript to take into account the work of [3].

- L194-197: "In this work … in real time". I think the content of these sentences is evident and doesn't need any further explanation.

We think that highlighting the importance of correcting in real-time and not a posteriori is relevant for the understanding.

- L210-211: "the angular accelerations … in the current setup". Can you estimate the error on Aerodynamic loads you are committing by neglecting these terms? Or alternatively how much Tin+Tgrav change by neglecting the angular accelerations?

The error committed can be judged as negligible from the following analysis. The plots below show, for a decay case in no-wind conditions, the time history of relevant quantities to assess the force correction. They are shown for Fx and My as they are the key contributions. In the manuscript, these plots about will be included and extended to all directions and more cases. The decay starts at about 10.5s from a displaced pitch position. The top plot shows the estimation of the aerodynamic force in x (surge) direction: blue solid line is the measured rotor force; the red solid line is the signal used to correct for inertial and gravitational contributions, i.e. the measured acceleration in x direction times the mass or the RNA; the green line is the aerodynamic force in x direction estimated by the model. As desired, the estimated aerodynamic force is approximately null (it is a no-wind condition where no aerodynamic force is expected on the rotor). The bottom plot shows the estimation of aerodynamic torque in y (pitch) direction about the s.w.l (tower bottom): the blue dashed line is the measured aerodynamic torque in y direction; the red solid line is the signal used to correct it, i.e. the measured acceleration in x direction times the mass or the RNA times the distance between the loadcell and the cog of the RNA; the blue solid line is the measured rotor force in x direction multiplied by the hub height (to calculate the torque at tower base); the red solid line is the signal used to correct it, i.e. the measured acceleration in x direction times the mass or the RNA times the hub height; the green line is the aerodynamic torque in y direction estimated by the model. Despite, in theory, the blue dashed contribution is not entirely corrected by the red dashed contribution, the two are almost equal. This shows that the missing correction term of RNA moment of inertia times the angular pitch acceleration is negligible, and overall, the estimated torque results in constant and null as desired. However, the uncorrected pitch moment could have a bigger effect in different motion cases, which is why the authors stated it as a drawback of the approach. Also, yaw direction remains totally uncorrected. Future developments will focus on how to estimate the angular accelerations relying on the acceleration sensors (as suggested by Reviewer 2).

[Figure]

[Figure]

- L213: "delays between sensors signals and hexapod feedback". This sentence is vague, and you should explain here that there is a delay (which is quantified afterwards) and explain which are the consequences of this delay.
  The sentence will be rephrased in the manuscript.

- L215-216: "de-synchronization of signals … acceleration inputs". I think the authors must explain more clearly why they didn't use filters. I suspect the first bending frequencies of the scale model (tower and blade FW) could be at around 10 Hz. The forces resulting from the vibrations of the scale model components are measured by the load and are transferred to the numerical model of the platform causing it to respond at the same frequency of the flexible vibrations. This normally gives rise to a loop that could make the HIL system unstable. Why it's not the case? Is it related to the motion tracking performance of the hexapod which maybe filters out and does not track motions at high frequency. I ask the author to provide the transfer function (in a plot or also by means of values at some frequencies reported in the text) of the hexapod setpoint-->actual movement in terms of magnitude and phase.
  It was decided not to use filters because all the filters and cutoff frequencies tested during the calibration of the system resulted in an unstable HIL system. Indeed, the first tower frequency is 12.5Hz, and the first blade (flapwise) frequency is around 16Hz. What most likely filters out the higher frequencies is the floater dynamic system model in combination with the motion tracking of the hexapod. The Bode plot of the Hexapod is shown above.

- L218: "This reduces errors … using a different sensor". The meaning of this sentence is not clear, rephrase it.
  The sentence will be rephrased in the manuscript.

- L259: "Prior to wind tunnel … the results". It seems a reasonable step and I don't see the need to give this advice to other people. You can simply say that The HIL setup was verified in cases without wind.
  The sentence will be changed accordingly in the manuscript.

- L268: "it corresponds to testing in Open-Loop". It doesn't. It corresponds only if the robot tracks perfectly the setpoint from the numerical model. However, the robot tracking performance is not proved anywhere. I suggest removing this part of the sentence.
  It refers to the fact that there is no feedback, and the simulation gives the same exact position output in standalone (i.e. on a non-real-time pc, where the tuning happened) and in open-loop (i.e. running on the real-time machine, at model scale, with the hexapod activated). The actual position differs from the setpoint and the tracking performance is shown above. It will be made clearer in the text.

- L269: "FAST". Can you be more accurate and specify which version of (Open)FAST you are using?

  We used FAST v8.16 for this work, as we had the model available in that version.

- L276: "stiffness". I get that you must adjust damping since HydroDyn uses a more complex model, but why you had to adjust stiffness? how was the mooring stiffness matrix obtained? How did you conduct the tuning? Are final values close to initial guesses or not? Is the difference expected?

  The mooring stiffness matrix is given in the specification document of the floater concept (https://doi.org/10.18419/DARUS-514). The overall stiffness was adjusted to match more closely the response of FAST and the Simulink model. The tuning is performed by repeating decay cases in all DOFs until a good match is found. Given that we tuned a linear model (at least for what concerns the mooring) with a nonlinear model (we used Moordyn in FAST), we chose an initial condition for the tuning. Inevitably, we expect differences with FAST if different initial conditions are used, or for wave cases.

- Figure 4: "at full-scale". In the results, report everything at full scale or at model scale.

  This will be modified throughout the manuscript

- L288: "neglecting the non-rotating rotor drag". I get what you mean, and I think it's better to say that the aerodynamic force developed by the non-rotating rotor is small.

  It will be modified accordingly in the manuscript.

- L294: "with the only exception of the yaw case". The yaw response isn't that bad.

  We agree that the yaw response doesn't seem bad; however, due to the absence of correction, all inertial loads in yaw do affect the response. It may be that this is more evident in other test cases. It was, anyway, important for us to stress the absence of correction and its (possible) effect. It will be modified in the manuscript, saying that the response in yaw shows a good match although there is no correction.

- L296-298: "This, as explained ... negligible effects". Rephrase, it's not clear. I think it's better to spend more line of text here to clearly explain why force correction on yaw is more critical than for the other degrees of freedom.

  The error we commit by not correcting the measured inertial torque in pitch and roll is small (estimated in a point above) because the force, which is corrected, is predominant in those DOFs due to the transport term (the hub height). Instead, in yaw, the inertial torque (not corrected) is not negligible, being the only component (no corrected force to transport). This explanation will be included in the text.

- L298-299: "Moreover, this study ... has a limited impact". I'd say that the study focuses on cases with aligned wind and waves, where the yaw response is expected to be small, thus the accuracy for this DOF can be lower than the others, especially surge, pitch and heave.

  It will be modified accordingly in the manuscript.

- L300: "the results obtained in the wind tests are accurate". This is not true. In these tests, the aerodynamic force resulting after correction is ideally zero. If it is introduced in the numerical model with a delay, it doesn't produce any error in the system response because it has zero magnitude. In these tests it was only possible to assess the accuracy of the force correction procedure but from its results it's not possible to say that tests with wind are accurate.

  If there was a (significant) delay in the numerical model (which accounts for both correction and dynamics in the same time step), we believe that any residuals from the force correction would have harmed the stability of the HIL system. This happened, for

example, when trying to introduce a filter for force and acceleration measurements. The reason why the filter option was, for the time being, discarded. Anyway, the text will be modified, stressing that the results assess the accuracy of the force correction procedure and can only give an indication of the accuracy of wind tests.

- Figure 6: I think this figure doesn't add much compared to figure 5 and can be removed.
the figure will be removed

- Section 3.3.3: I think this section should be used to show the FRF motion setpoint-->actual motion
See above.

- L302: "within a limit". Can you quantify this limit?
A "hard" limit is given by the stability of the HIL itself, and in keeping the delay small with respect to the dynamics one wants to study. There is no quantifiable answer for this; thus, the text will be modified.

- L305: "sensibly higher for the force sensor". How much higher?
The bandwidth of the force sensor is not reported in the specification. The resonant frequency is around 5000Hz, and this helps in understanding the usable frequency, which is anyway higher than the accelerometer one.

- L330-331: rephrase the sentence "The thrust force ... in the direction of thrust" because it's not clear.
It will be rephrased in the manuscript.

- L335: "where the part of the rotor ... causing the damping effect". This explanation is difficult to understand especially when it says "the opposite happens simultaneously on the other side, causing the damping effect.". Please rephrase it and make it more robust and less speculative.
It will be rephrased in the manuscript.

- L350: "this is also the region where the effects on wind are visible". The figure only shows the response with wind but not the case of waves without wind (like in Fig. 7-8) so it's impossible to tell at which frequencies the aerodynamic loads have a meaningful effect.
Unfortunately, no wave case in no-wind conditions was performed in that wind tunnel campaign, and so no comparison can be shown in the manuscript. However, it is known that the wind effects are in the low-frequency region. Although not suitable to be included in the paper (reasons follow), here we can show the comparison between HIL (closed loop) wave cases in wind and no-wind conditions preliminary tested in a previous campaign (unpublished). The floater concept is still the TripleSpar and also the sea state (Hs 3.04, Tp 9.5), but the floater dynamic model has few differences with the current one, and the HIL runs at 3 DOF only. The most visible effect is that the cases with wind and rotor operating have a lower peak at the pitch natural frequency for pitch DOF. However, to give the article a different cut and concentrate on the development of the HIL, it was decided not to deepen this analysis in the study. Future studies will focus

on the aerodynamic effect on FOWT dynamics making use of the setup.

[Figure]

- L355-356: "The dynamic response to wind and waves … this kind of study". This sentence is very broad. You must add reference of the literature that you mention and explain how your findings compare to theirs. Even better, you should add to figures which is the expected response (e.g., from numerical models).
  In the manuscript, a qualitative comparison of our findings to what is expected from other HIL studies will be added, in particular from [3] (this study is more comparable to our because the wind turbine was also operated at fixed speed and pitch). A qualitative comparison with the literature on numerical (engineering) models will be included. However, no numerical study was conducted on purpose to compare with the wind and wave results of our setup, and this is motivated by the focus rather on the setup development than on studying a FOWT concept using the setup. The objective of this paragraph remains proving that the setup has the capability of reproducing wind and wave conditions and can be used for that in future studies.
- Figure 9: Can you do some averaging when computing spectra (e.g., with Pwelch), otherwise it's difficult to identify the features of the spectra.

The figure will be changed with spectra computed with Welch's method:

[Figure]

- L359-360: "only one wind tunnel HIL setup exists". Not true. These setups are scarce, but there are others, for example [4]. As the authors acknowledge, the published work on this kind of setups is not much, so I suggest them to extend their survey of literature that now is a bit limited.
  The sentence will be changed in the text, and the literature review will be enriched in the introduction, also including [4].

- L368-370: "By measuring forces at the tower top... force ratio". Not true, this is done in [3]. The novelty here is using measured accelerations.
  It will be changed in the text

- L373: "unsteady aerodynamic phenomena". I invite you to caution about these phenomena you mention because it has not been clearly proven yet that they exist and how the HIL setup presented in this work may eventually contribute to their understanding.
  It refers to the findings of a previous study by the authors (https://doi.org/10.5194/wes-9-343-2024). The text will be modified, however.

- L374-377: "the versatility of ... aerodynamic phenomena". These sentences are broad and a bit useless because the flexibility of use mentioned by the authors is not demonstrated in the article. I suggest recalling the main results of the article instead.
  This part will be changed accordingly in the manuscript

- L378-379: "A preliminary study ... degrees of freedom". Be more accurate on what you found.
  It will be expanded in the text

- L393: what you mean with "theoretically measured"?
  It refers to the analytical formulation of what forces the load cell is sensing, with the simplifications of (A1) (e.g. rigid bodies). It will be changed in the text.

- L405: where it has been "already calculated"?

  The calculations are in (A1). It will be made clearer in the text.

**Technical corrections**

Technical corrections will be implemented in the manuscript (unless otherwise specified).

- Title: I suggest removing "On the".

- Abstract: replace "met-ocean conditions" with "wind and wave excitation".

- L11: remove "dynamic".

- L12: "hexapod" it is said before that the turbine is mounted on a hexapod robot so there is no need to specify it again.

- L20: remove "deeper"

- L24: "FOWTs experience (rigid-body) motions".

- L61: remove the comma after "rotor".

- L106: "a velocity scale of 3". It should be "1:3" to be consistent with the geometry scale reported at line 105.

  "1:148 model" is equivalent to "with a length scale of 148".

- L106: replaced "scaled" with "designed".

- L127: replace "coupling" with "coupler".

- L131: replace "controller" with "computer" or "machine".

- L131: replace "perform the hardware in the loop" with "run the hardware-in-the-loop" controller.

- L132-133: "(DAQ)" and "(HMI)" These acronyms are not used anywhere else in the text so I think it is unnecessary to define them.

- L134: "real-time machine".

- L155: remove brackets and add "whose scaling is governed by" after "motion frequency".

- L181: replace ". This" with "; this".

- L182: add "between the physical and numerical parts of the experiment." after "mismatch".

- L183: rephrase the sentence "due to manufacturing ... the mass scale" because it is hard to read.

- L187: replace "read" with "measured".

- L187: "the measurements are heavily compromised". I would say that is not possible to use the measurements as a feedback.

- L232: what does "derived" mean here?

- L234: what does "flexibly targeted" mean?

- L235: remove "The general equation of motion ... hereafter" because it's a repetition of the sentence afterwards.

- L238: I suggest removing "R2 is the quadratic damping (here not modelled)" and remove R2 from Eq. 4 to avoid confusion.

- L240: I suggest removing "Fmoor is the mooring load (here not modelled)" and remove Fmoor from Eq. 4 to avoid confusion.

- L242: remove "The viscous effects ... damping matrices".

- L252: remove "(SS_Fitting)"

- L253: replace "the radiation calculation" with "$F_{hydro}$"

- L262: replace "3" with "three".

- L264: "*loop*" instead of using italics recall what is the loop.

- L273: remove "The wave response ... in wave conditions".

- L320: replace "in wind conditions" with "with wind blowing".

- L323: "at rated conditions". Can you recall the wind speed and rotor speed?

- L331: replace "no control is present" with "rotor speed and blade pitch are fixed".

- L358: remove "(HIL)".

- L359: remove "(FOWTs)".

- L366: add "Because of the additional complexity introduced by closing the loop," before "Investigating the effect...".

- L367: remove "therefore".

- L415: "3" is it "Eq. 3"?

- Appendix A2: I suggest replacing "rotating frame" with "moving frame". In wind turbine, "rotating frame" is usually used to identify the coordinate system that rotates together with the rotor.

- L426: capital R in "Rotation" should be small.

- L433 and 434: replace "rotating" with "rotated".

**References**
Suggested references will be included in the manuscript.

[1] Papi, F., Jonkman, J., Robertson, A., and Bianchini, A.: Going beyond BEM with BEM: an insight into dynamic inflow effects on floating wind turbines, Wind Energ. Sci., 9, 1069–1088, https://doi.org/10.5194/wes-9-1069-2024, 2024.

[2] Schulz, C. W., Netzband, S., Özinan, U., Cheng, P. W., and Abdel-Maksoud, M.: Wind turbine rotors in surge motion: new insights into unsteady aerodynamics of floating offshore wind turbines (FOWTs) from experiments and simulations, Wind Energ. Sci., 9, 665–695, https://doi.org/10.5194/wes-9-665-2024, 2024.

[3] Fontanella, A., Facchinetti, A., Daka, E., and Belloli, M.: Modeling the coupled aero-hydro-servo-dynamic response of 15 MW floating wind turbines with wind tunnel hardware in the loop. Renewable Energy, Volume 219, Part 1. https://doi.org/10.1016/j.renene.2023.119442, 2023.

[4] Jiang, Z., Wen, B., Chen, G., Tian, X., Li, J., Ouyang, D., Peng, Z., Dong, Y., and Zhou, G.: Real-time hybrid test method for floating wind turbines: Focusing on the aerodynamic load identification. Journal of Ocean Engineering and Science. https://doi.org/10.1016/j.joes.2024.06.002, 2024.

Dear Authors,

It was very interesting to review this paper. Empirical data plays a key role in increasing our understanding of the physics of such complex systems, and for the development of numerical tools.

The paper is well written and well structured, and there is no doubt that a considerable amount of work has been put in the development of a HIL system for testing of offshore wind turbines in a wind tunnel.

The introduction states that this work primarily focuses on the verification and validation of the setup. This is an important step in the development of the setup, but one would expect more results and a more thorough analysis of the results to be really convinced about the verification and validation of the setup.

More analyses on the verification of the setup have been performed following feedback and will be added to the manuscript. This includes an analysis of the force correction showing the measured and estimated loads and estimating the error introduced by non correcting the rotational accelerations, and the estimation of the transfer function between motion command and actuation.

Also, the introduction is interesting but lacks clarity regarding the advantages and the limitations of model testing in a wave basin compared to a wind tunnel. In the abstract , it is stated that the wind tunnel HIL tests overcome the scaling issues of traditional wave basin tests. I would think that this is false, and that the only advantage of wind tunnel tests compared to wave basin tests is in the quality of the wind. Also, additional challenges arise for the wind tunnel tests due to the non-froude scaling and the very limited mass of the RNA.

We agree that the main advantage is the wind flow quality. However, given models of the same dimensions, in fully-physical wave basin tests the Reynolds mismatch will be higher due to the compulsory Froude scaling, with a greater effect on the aerodynamic performance of the rotor. Hybrid tests in wind tunnel allow for a smaller Reynolds mismatch. And we also agree that new challenges are introduced with hybrid testing in wind tunnel. The introduction will be modified to be clearer on possible advantages and challenges, and the statement in the abstract will be changed to make it less affirmative.

The paper in its currents state needs more work but will be worth publishing once the two above comments have been addressed. The specific comments below are meant to help you to improve the paper regarding my two comments above, but additional results and discussions are needed regarding the validation and verification.

**Specific comments:**

L38-40: One of the main advantages of laboratory testing compared to field testing is that the environment is controlled in a laboratory. I think that this should be stated as one of the reasons why laboratory testing is often preferred.

It will be added in the text

L51-55: It is important to give a clear review of basin tests with performance scaling.

- Why do you state that an arbitrary velocity scaling is used while my impression is that Froude scaling is often used. See for example Bredmose (2017). By using Froude scaling in basin tests, the problem about mismatch in the scaling of the gravity is for example avoided.

  **The mistake will be corrected in the text**

- How do you define low quality wind flow. The wind quality is the main difference between basin tests and wind tunnel tests and deserves therefore more than one sentence.

  **It refers to a quality not comparable to the one achievable in a wind tunnel, e.g. in terms of flow uniformity, turbulence intensity, but also temperature control, etc. This part will be expanded in the text.**

- The mass of the RNA is difficult to match, but is it really not achievable? Already in 2017, Bredmose was only 12% above the specification. But even with a RNA that is a bit heavier than desired, it is still possible to achieve the correct centre of gravity, which is correct for rigid models.

  **It will be modified accordingly in the text.**

L95: The hybrid setup: Can you add more details about the quality of the flow in the wind tunnel (spatial and temporal variation, capabilities, ...), since this is supposed to be better for wind tunnels compared to basins. And this is one of the main "selling" arguments for HIL wind tunnel tests.

The wind tunnel has an open-jet closed-circuit configuration. The octagonal nozzle has dimensions of 2.85m x 2.85m (equivalent diameter of 3m) and a contraction ratio of 3:1. The flow is uniform with approximately 0.5% turbulent intensity at 1m from the jet exit (where the current model was placed) and lower than 2% at 6m from it. The nozzle opens on a 13m long and about 6m wide and 6m high test section. The uniform-flow region reduces at 6 m from the jet exit from 3×3 m2 to 2×2 m2. The tunnel is driven by a fan with an electrical engine of 500 kW, and the temperature is kept constant by a heat exchanger, which provides up to 350 kW of cooling. The maximum wind speed is 35m/s. More specifications of the tunnel will be included in the manuscript. A characterisation can be found in [https://doi.org/10.4233/uuid:057fa33f-82a3-4139-beb8-53f184cd1d57] in section 2.2.5.

L117: The hexapod: Since the main objective of the paper is to verify and validate the setup, one expects more information about the tracking errors of the hexapod. "... without tracking errors": Does this mean 0 errors, or negligible? A bode plot with amplitude and phase would be very valuable here.

It means: without significant tracking error. The Bode plot is reported here and will be added in the

manuscript; see the answer to Reviewer 1 for more details.

[Figure]

L143: I do not agree with the statement that the main driver for hybrid tests in wind tunnels is to overcome the Fn-Rn scale conflict. The same issue is present in HIL wind tunnel tests and therefore, performance scaling is also used in wind tunnels. The main driver for HIL wind tunnel tests is in the quality of the wind. It is also important to highlight the challenge that arises in HIL wind tunnel tests, due to the non-Froude scaling. While this is not a problem in basin since Fn scaling is used there in combination with performance scaling.
See answer above. We agree that hybrid testing doesn't overcome Fr-Re conflict. However, given the same model dimension, it can reduce the Re mismatch resulting in better aerodynamic performance of the rotor. The manuscript will be modified with a clearer explanation of what is the main advantage (wind quality), what are possible advantages (less Re mismatch, but still need of performance scaling), and disadvantages (RNA mass scaling, force correction needed, ...).

L189: Please explain how you arrive at the mismatch in ratio of 150. The mismatch in ratio between the aero and the inertial forces is the same as the Renolds mismatch (given in table 2) times the mass mismatch (10). The mismatch in ration between aero and gravitational is the same as the mismatch in Rn*Fn*mass mismatch (10).
The aerodynamic forces are correctly scaled (according to the arbitrary scale factor_length=148 and factor_velocity=3) because the rotor is performance-scaled. The acceleration mismatch (being non-Froude) of about 15 together with the mass mismatch of 10, makes the inertial and gravitational loads about 150 times bigger than if they were correctly (Froude) scaled. These numbers are shown to highlight the importance and challenge of estimating the (small) aerodynamic part out of the measured loads.

L221: Floater Dynamics. Since the objective of the paper is verification and validation of the HIL setup, this section about the numerical model should be exhaustive. The information is scarce, and it is difficult to know what is exactly included in the numerical model.
The section about the floater modelling will be expanded in the manuscript. However, we would like to clarify that the floater numerical modelling is simplified here and will be the object of future work to explore, either with physics-based or data-driven approaches. This work intends to be

more centred on the HIL architecture and its verification and capabilities, for which a numerical model computing a realistic dynamic response is needed, but the model itself is not the focus. This also explains why we opted for the tuning approach, and we didn't show much about the results of the model, in addition to its natural frequencies and damping.

- Equation 4: Added mass on the LHS is a part of the F_Hydro. Same for F_rad in L454.
  Corrected in the manuscript

- L246: Is it only radiation damping or should it be radiation loads?
  Corrected in the manuscript

- L247: Please clarify what is meant by wave diffraction since Faltinsen and Newman have different interpretation for this term.
  Wave diffraction force is used in the sense of Newman (1977), i.e. the total first-order wave excitation on a fixed body, including both the Froude–Krylov and scattering components. It will be specified in the text.

- What about second order wave loads?
  Second-order wave loads are not modelled in this work. They are included in equation A7 in appendix to make it more generally applicable.

- L456: What is F_diff,2 and why is it not included in A7.
  It is a mistake; it will be corrected.

L266: Floater simulator: If the objective is to verify and validate, then one should compare FAST and Simulink with the same parameters and verify for a good agreement. By tuning the parameters for agreement, you are hiding possible errors in the numerical model.
We agree that comparison using the same parameters is the standard approach. The discrepancies between the FAST and Simulink models that led to the tuning were likely due to the use of more advanced hydrodynamics in HydroDyn and the dynamic mooring model in MoorDyn. The latter needed to be approximated by a linear mooring model in our Simulink implementation, but it's kept dynamic in FAST. Given these differences, we chose a tuning approach to match the rigid body frequency and damping of the benchmark model (FAST) for selected test cases, i.e. the decay tests we then used for assessing the HIL chain first and estimating the aerodynamic damping after. This allowed us to replicate the key dynamic characteristics of the system better than using the input parameters from the definition document.

Figure5: The agreement in pitch is not so good while pitch is one of the key DOF for Floating wind turbines. Would it be possible to use the angular pitch acceleration derived from the accelerometers at tower base and top?
Using the second accelerometer to derive the angular accelerations would be a good option that we will explore in future HIL modelling and tested in next campaigns. To show more about the verification of the force correction, and the entity of the error introduced in pitch and roll DOF, figures showing the measured and estimated forces will be added to the manuscript.

L309: What do you do with the estimated latency? Can you give additional explanations on why you believe that you can split the delay evenly between motion and communication. Other laboratories have tried to compensate the delay, see for example [2] Have you tried similar?
The split of the delay in half was just a guess. We assumed that the communication between the real-time machine and the hexapod (position command: digital-analogue conversion, analogue signal, analogue-digital conversion, inverse kinematics computation) took the same time as the communication between hexapod and real-time machine(position actual: direct kinematics computation, digital-analogue conversion, analogue signal, analogue-digital conversion). The

delay induced by the hexapod actuators themselves was neglected in the splitting. The value given in the manuscript was, however, rounded up, and we believe it still gives a good order of magnitude estimate. In the present work, no delay compensation was performed.

**Technical Corrections:**

L34: Can you add a reference to the statement about premature maintenance.
Since there is no scientific reference available, but only newspaper articles can be found on the occurrence (https://www.windtech-international.com/windtech-future/the-case-of-hywind-farm-wind-turbines) , the sentence will be deleted.

L49: Rn does not compromise the hydrodynamic loads (viscous effects are correctly scaled), but it compromises the gravity related hydrodynamic loads.
This will be modified accordingly in the text

L51: Is it correct to state that an arbitrary velocity scaling factor is used in performance scaled tests? My impression is that in basin tests, performance scaling is used in combination with Fn scaling. See for example [1].
The mistake will be corrected in the text.

Figure2: Did you have to adjust the blade pitch angle to achieve the desired thrust?
No, the blade pitch is fixed at 0deg. The rotor is made in a single piece and it is not possible to change the pitch. Above rated operating points cannot be tested with this rotor.

L146: Even with non-Froude, inertial phenomena can be correctly reproduced. The problem is that the balance between inertial and gravitational loads is not correctly reproduced.
This will be modified accordingly in the text

Table 3: Please add Froude mismatch
It will be added to the table in the manuscript

L184: The same challenge would arise also for custom made components. The challenge is mainly due to the complexity of RNA and the need for sensors and actuators.
This will be modified accordingly in the text

References
Suggested references will be included in the manuscript.

[1] H. Bredmose *et al.*, "The Triple Spar campaign: Model tests of a 10MW floating wind turbine with waves, wind and pitch control," *Energy Procedia*, vol. 137, pp. 58–76, Oct. 2017, doi: 10.1016/j.egypro.2017.10.334.

[2] Zhihao Jiang, Binrong Wen, Gang Chen, Xinliang Tian, Jun Li, Danxue Ouyang, Zhike Peng, Yehong Dong, Guiyong Zhou, Real-time hybrid test method for floating wind turbines: Focusing on the aerodynamic load identification, Journal of Ocean Engineering and Science, 2024,